META-RESEARCH ARTICLE

# Analysis of 567,758 randomized controlled trials published over 30 years reveals trends in phrases used to discuss results that do not reach statistical significance

**Willem M. Otte**[1,2], **Christiaan H. Vinkers**[3], **Philippe C. Habets**[3], **David G. P. van IJzendoorn**[4], **Joeri K. Tijdink**[5,6]*

1 Biomedical MR Imaging and Spectroscopy, Center for Image Sciences, University Medical Center Utrecht, Utrecht, the Netherlands, 2 Department of Child Neurology, UMC Utrecht Brain Center, University Medical Center Utrecht, Utrecht, the Netherlands, 3 Department of Psychiatry, Department of Anatomy and Neurosciences, Amsterdam UMC, Amsterdam, the Netherlands, 4 Department of Pathology, Stanford University School of Medicine, Stanford, California, United States of America, 5 Department of Ethics, Law and Humanities, Amsterdam UMC, Amsterdam, the Netherlands, 6 Department of Philosophy, Vrije Universiteit, Amsterdam, the Netherlands

* j.tijdink@amsterdamumc.nl

**Data Availability Statement:** All used PubMed IDs, detected phrases, co-text extractions, manually identified P values, and processing scripts are

## Abstract

The power of language to modify the reader's perception of interpreting biomedical results cannot be underestimated. Misreporting and misinterpretation are pressing problems in randomized controlled trials (RCT) output. This may be partially related to the statistical significance paradigm used in clinical trials centered around a $P$ value below 0.05 cutoff. Strict use of this $P$ value may lead to strategies of clinical researchers to describe their clinical results with $P$ values approaching but not reaching the threshold to be "almost significant." The question is how phrases expressing nonsignificant results have been reported in RCTs over the past 30 years. To this end, we conducted a quantitative analysis of English full texts containing 567,758 RCTs recorded in PubMed between 1990 and 2020 (81.5% of all published RCTs in PubMed). We determined the exact presence of 505 predefined phrases denoting results that approach but do not cross the line of formal statistical significance ($P < 0.05$). We modeled temporal trends in phrase data with Bayesian linear regression. Evidence for temporal change was obtained through Bayes factor (BF) analysis. In a randomly sampled subset, the associated $P$ values were manually extracted. We identified 61,741 phrases in 49,134 RCTs indicating almost significant results (8.65%; 95% confidence interval (CI): 8.58% to 8.73%). The overall prevalence of these phrases remained stable over time, with the most prevalent phrases being "marginally significant" (in 7,735 RCTs), "all but significant" (7,015), "a nonsignificant trend" (3,442), "failed to reach statistical significance" (2,578), and "a strong trend" (1,700). The strongest evidence for an increased temporal prevalence was found for "a numerical trend," "a positive trend," "an increasing trend," and "nominally significant." In contrast, the phrases "all but significant," "approaches statistical significance," "did not quite reach statistical significance," "difference was apparent," "failed to reach statistical significance," and "not quite significant" decreased over time. In a random

openly shared at: https://github.com/wmotte/almost_significant (v1.0; http://doi.org/10.5281/zenodo.4313162). and here: https://github.com/wmotte/almost_significant/tree/main/Fig_and_Data.

**Funding:** The authors received no specific funding for this work.

**Competing interests:** The authors have declared that no competing interests exist.

**Abbreviations:** BF, Bayes factor; CI, confidence interval; PDF, portable document format; RCT, randomized controlled trial.

sampled subset of 29,000 phrases, the manually identified and corresponding 11,926 $P$ values, 68,1% ranged between 0.05 and 0.15 (CI: 67. to 69.0; median 0.06). Our results show that RCT reports regularly contain specific phrases describing marginally nonsignificant results to report $P$ values close to but above the dominant 0.05 cutoff. The fact that the prevalence of the phrases remained stable over time indicates that this practice of broadly interpreting $P$ values close to a predefined threshold remains prevalent. To enhance responsible and transparent interpretation of RCT results, researchers, clinicians, reviewers, and editors may reduce the focus on formal statistical significance thresholds and stimulate reporting of $P$ values with corresponding effect sizes and CIs and focus on the clinical relevance of the statistical difference found in RCTs.

## Introduction

Individual clinical researchers are subject to the mythical heritage or paradigm of the peculiar and well-recognized 0.05 significance threshold ($P = 0.05$, alpha) that claims that findings below this predefined value reflect a true finding. In contrast, $P$ values not making the cut indicate no effect (null hypothesis not rejected). Consequently, individuals submitting randomized controlled trial (RCT) publications often dance the "significance dance" to describe outcomes around the 5% alpha level. One of the challenges is that an overreliance on one fixed cutoff is that to "find" that a treatment works or not, $P$ values below 0.05 are often thought to be mandatory. However, the $P < 0.05$ threshold is a simple rule to reject the null hypothesis, controlling for type I and II errors. This has led to misinterpretations that dichotomized the $P$ value ($P < 0.05$ = true effect, $P > 0.05$ = no effect).

Interestingly, the vast majority (96%) of biomedical articles report $P$ values of 0.05 or less [1,2]. Unseen, but behind this peculiar distribution of published $P$ values are those that did not make it below 0.05. In psychology, the occurrence of reporting $P$ values between 0.05 and 0.1 —about 40%—is relatively high [3]. Less is known about these numbers in clinical research. In a small sample of 722 articles in oncology research, 63 articles (8.7%) used trend statements to describe statistically nonsignificant results [4].

Authors could misrepresent nonsignificant trial results through biased emphasis or phrasing of the outcomes. A well-known example of this so-called "spin" practice is switching the emphasis from nonsignificant primary to significant secondary outcomes. This highlighting favorable results while suppressing unfavorable data is considered misrepresentation [5]. Another example is the use of linguistic spin. Linguistic spin could distort the interpretation of trial results in reframing or modifying the reader's perception into a beneficial interpretation despite a statistically nonsignificant difference in the primary outcome [6].

And finally, recent reports indicate a high percentage (ranging from 47% to 66%) of detected spin across medical disciplines [7–10].

Strong preferences for $P$ values below 0.05 may also lead to creative linguistic solutions. Reporting nonsignificant results as essential or noteworthy findings may effectively invite scholars to overstate their findings and present uncertain, insufficient evidence (e.g., with a high risk of bias or other methodological weaknesses) as "breakthrough" research with clear clinical impact. These linguistic trends have possibly a temporal element. Some language phrases will be more successful in convincing editors and reviewers over time than others. Given the relatively rules-oriented RCT research environment, we expected creative linguistics

regarding significance phrases in published RCTs and trends over time for the most favorite phrases.

Insight in this practice is essential as the success of an RCT is partly determined by the way the results are presented in a manuscript [11]. Effective interventions and procedures with clear and significant outcomes that promise to improve patient care will most likely guide acceptance decisions. However, in papers without apparent clinical breakthroughs, the language used to highlight potential beneficial treatments may nonetheless convince reviewers and readers [12,13]. Also, for RCTs, the cornerstone of evidence-based medicine, 2 independent studies have detected that positive reporting and interpretation of primary outcomes in RCTs were frequently based on nonsignificant results [14,15]. Persuasive phrasing like "marginally significant" and "a trend toward significance" may disguise nonsignificant results. Given that there is essentially no clinically relevant distinction between a type I error of 4%, 5%, or 6%, it is interesting to understand how the formulations regarding $P$ values just above 0.05 change over time.

Therefore, the study aims to detect the prevalence of specific nonsignificant phrases in RCTs and determine what phrases correspond with the reported statistical nonsignificant findings to explore potential consequences that arise with the significant threshold of $P < 0.05$. We do this by quantitatively analyzing RCT full texts registered in the last 3 decades in the PubMed database. We determined the use of 505 most common phrases describing nonsignificant results and characterized the trends over time. In a subset, we manually assessed their associated $P$ values. We expected to find similar percentages of phrases associated with nonsignificant results in RCTs as reported in other (mostly nonclinical) studies [14,15]. We also hypothesized to detect changes in phrase prevalences over time, assuming continuous evolution of phrasing in reporting of nonsignificant RCT results. Finally, we anticipated that the phrase-associated $P$ values would predominantly be associated with a $P$ value in the range of 0.05 to 0.15.

## Methods

### Selection of RCTs

A flowchart shows the consecutive processing steps (Fig 1). We identified all RCTs in the PubMed database and excluded animal studies, non-English studies, and studies that were not actual RCT reports [September 20, 2020] with the following query: "All[SB] AND Humans [MESH] AND English[LANG] AND Clinical Trial[PTYP] NOT protocol[TITLE]." Our custom search query is not previously validated. However, PubMed's internal "Clinical trial" filter is characterized with an average sensitivity of 87.3%, specificity of 34.8%, and precision of 54.7% [16]. It is optimized for a sensitive and broad rather than a specific and narrow yield. In comparison, a clinical query optimized for sensitivity reached 92.7% sensitivity and 16.1% specificity, whereas a clinical query optimized for specificity reached a sensitivity of 78.2% and a sensitivity of 52.0%. Our query is thus a compromise between sensitivity and specificity. We expect that further restricting the query to "Humans," "English," and no protocols will have increased our specificity [16].

Subsequently, we collected the portable document format (PDF) for all available RCTs across publishers in journals covered by our institution's library subscription. All trial PDFs were converted to XML and subsequently plain text in XML format using publicly available Grobid software (v. 0.6.2). We converted the plain text to lower case and removed diacritical symbols as there are various types of quote Unicode characters. Subsequently, we searched for exact matches (i.e., grep command line tool) of the predefined phrases in this cleaned text.

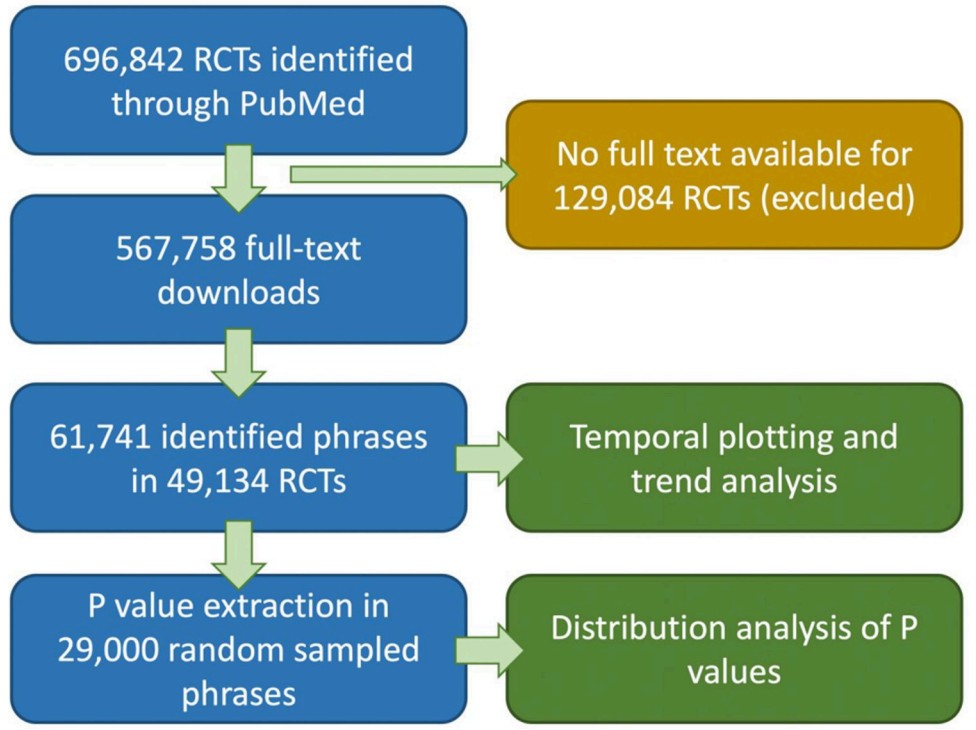

**Fig 1. Flowchart of the study processing pipeline.** RCT, randomized controlled trial.

## Phrases

We predefined 505 phrases potentially associated with reporting nonsignificant results (**S1 Table**). We used a list provided by statistician Dr. Matthew Hankins on his "Still not significant" blog, based on actual examples found in the biomedical and psychology literature [17,18].

## Prevalences

We restricted the publication time frame to 3 decades: January 1990 to September 2020. The total phrase-positive RCT prevalence was determined for each publication year. To increase the temporal robustness of individual phrase prevalence estimations, we binned RCTs according to their publication date into periods of 3 years. For each phrase detected as an exact match in the full texts, time period prevalences were calculated by dividing the number of RCTs that included one of the 505 phrases describing nonsignificant results by the total number of RCTs within that period. The 95% confidence intervals (CIs) were determined with Yates continuity correction [19].

## Statistical analysis

To obtain evidence on phrase changes over time, we used a Bayesian linear regression [20] and determined Bayes factors (BFs) for each fitted model. This ratio measure determines the relative evidence of a model with a linear slope in the temporal prevalence data over a null model with an intercept only. For example, a BF of 5.0 means that the prevalence of a specific significance phrase over time is 5 times more probable with a linear change over time than with no linear change over time. Nonetheless, multiple suggestions for interpreting BF divisions are

available. A commonly used list divides the evidence into 4 strength ranges: BF between 1 and 3.2 are "not worth more than a bare mention," between 3.2 and 10 are "substantial," between 10 and 100 are "strong," and >100 are "decisive" evidence [21]. To our knowledge, there is no evidence that reporting BFs is also subject to suspicious phrasing. We used the R package "BayesFactor" for statistical analysis. Model priors were uninformative.

### Associated *P* values

Phrases may refer to *P* values in broadly 2 types: a direct referral, with the corresponding *P* value, directly followed after the phrase, mostly in parentheses (e.g., "The drug effect was almost significantly lower in group B (*P* = 0.052)"). The other type often found in Discussion sections, typically contains longer range referrals to previously mentioned results, displayed in figures and tables. We tried to quantify the first type of referral by manually extracting the *P* value within the first 100 characters directly following the extracted phrases within 29,000 random sampled phrases for the full set of phrases. This sample size was achieved through distributed labeling with all authors independently extracting *P* values from 5,000 to 7,000 sentences. We evaluated our interrater *P* value variability based on 50 sentences shown to 2 raters but mixed within the larger set of extractions and expressed as the mismatch percentage.

## Results

We obtained the full text of 567,758 full texts of the total of 696,842 PubMed-registered RCTs (81.47%) (**Fig 1**). From the 505 predefined significance phrases, 272 were present in the full-text corpus at least 1 time. In total, 49,134 RCTs within the 567,758 full texts had a full-text match (61,741 phrases). The yearly prevalences are shown in **Fig 2**. The overall phrase-positive RCT prevalence was 8.65% (95% CI: 8.58% to 8.73%), and this percentage was stable over time.

The number of detected RCTs with phrases associated with reporting of nonsignificant results were unequally distributed (**Table 1**). The most prevalent phrases were "marginally significant" (present in 7,735 RCTs), "all but significant" (7,015 RCTs), "a nonsignificant trend" (3,442 RCTs), "failed to reach statistical significance" (2,578 RCTs), and "a strong trend" (1,700 RCTs).

We found evidence for a temporal change in multiple prevalences (**S2 Table**). From the phrases with a BF above 100 the RCT, prevalence increased from 0.005% to 0.05% ("a numerical trend"), 0.098% to 0.23% ("a positive trend"), 0.067% to 0.346% ("an increasing trend"), and 0.036% to 0.201% ("nominally significant"). Whereas the phrases—"all but significant," "approaches statistical significance," "did not quite reach statistical significance," "difference was apparent," "failed to reach statistical significance," and "not quite significant"—sharply decreased over time (**Fig 3**). An additional 17 phrases had "strong" BFs between 10 and 100 (**S1 Fig**). A total of 15 phrases had a BF between 3.2 and 10 (**S2 Table**), indicating "substantial" evidence for a temporal change. The remaining phrases are "not worth more than a bare mention."

### Associated *P* values

Within the random sample of 29,000 RCTs containing one of the nonsignificant phrases, we extracted 11,926 *P* values (41.1%) within the "100 characters" range. Interrater *P* value variability, based on a sample of 50 similar extractions—hidden within the larger random sample and seen by 2 authors—was less than 4%.

The *P* value distribution was characterized with a high prevalence within the 0.05 to 0.15 range with a median of 0.06. In the distribution of all detected *P* values, we found the 25% to

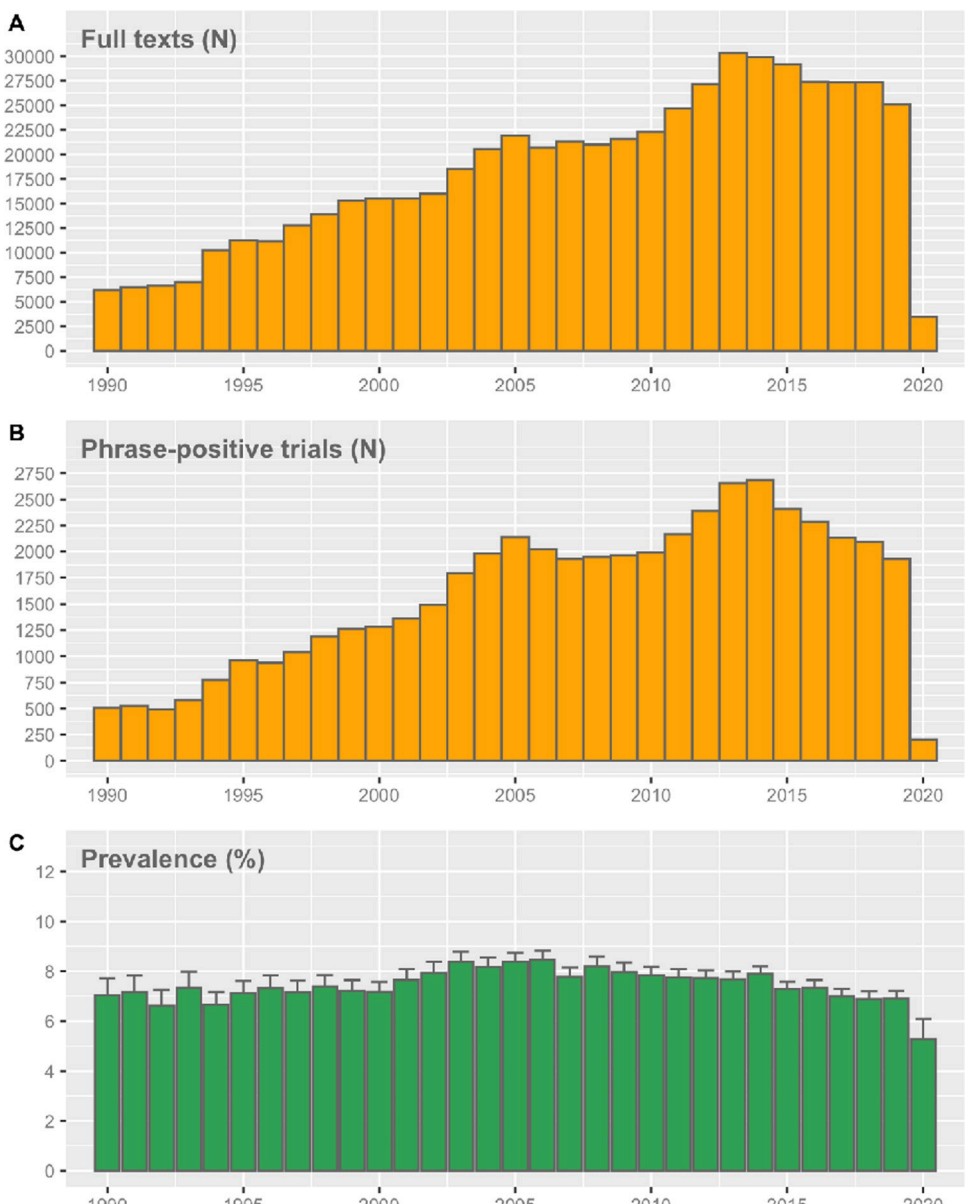

**Fig 2.** The number of analyzed full texts (**A**), number of phrase-positive RCTs (**B**), and the corresponding prevalence (**C**) over time. Error bars represent the 95% CI. The underlying data can be found here: https://github.com/wmotte/almost_significant/tree/main/Fig_and_Data. CI, confidence interval; RCT, randomized controlled trial.

75% interval $P$ values between 0.05 and 0.08. The 5% to 95% interval had $P$ values between 0.006 and 0.15 (see **Fig 4**). The proportions of $P$ values as being categorized as $<0.05$, between 0.05 and 0.15, or above 0.15 are given in **S3 Table**.

Some phrases were highly associated with a $P$ value between 0.05 and 0.15 (**Fig 5**, **S2 Fig**). The highest percentages of the following frequent phrases were found in this particular range of 0.05 to 0.15 for "almost reached statistical significance," "almost significant," "a strong trend," "did not quite reach statistical significance," "just failed to reach statistical significance," "near significance," and "not quite significant" (**Fig 5**).

**Table 1. The identified number of phrases (frequency *n* > 100).**

| Phrase | Total RCTs |
| --- | --- |
| Marginally significant | 7,735 |
| All but significant | 7,015 |
| A nonsignificant trend | 3,442 |
| Failed to reach statistical significance | 2,578 |
| A strong trend | 1,700 |
| Nearly significant | 1,391 |
| A clear trend | 1,372 |
| An increasing trend | 1,202 |
| Only marginally significant | 1,149 |
| A significant trend | 1,124 |
| Potentially significant | 1,104 |
| Significant tendency | 1,064 |
| A positive trend | 1,055 |
| A decreasing trend | 962 |
| Marginal significance | 887 |
| A slight trend | 885 |
| Almost significant | 813 |
| A statistical trend | 811 |
| Approaching significance | 796 |
| Nominally significant | 740 |
| Quite significant | 547 |
| Near significant | 546 |
| An overall trend | 445 |
| Likely to be significant | 425 |
| Difference was apparent | 409 |
| Uncertain significance | 383 |
| Did not quite reach statistical significance | 379 |
| A weak trend | 343 |
| Marginally statistically significant | 314 |
| Tended to be significant | 293 |
| Possible significance | 286 |
| Not quite significant | 266 |
| A favorable trend | 261 |
| Just failed to reach statistical significance | 252 |
| A negative trend | 225 |
| Almost reached statistical significance | 219 |
| A possible trend | 218 |
| Fell short of significance | 214 |
| Not as significant | 204 |
| A small trend | 185 |
| A numerical trend | 184 |
| Slightly significant | 182 |
| Reached borderline significance | 165 |
| Near significance | 156 |
| Weakly significant | 147 |
| Moderately significant | 146 |
| An apparent trend | 145 |

(*Continued*)

**Table 1.** (Continued)

| Phrase | Total RCTs |
|---|---|
| Barely significant | 135 |
| Practically significant | 135 |
| A definite trend | 131 |
| An interesting trend | 129 |
| Almost statistically significant | 126 |
| Marginally nonsignificant | 101 |
| Possibly significant | 100 |
| Significantly significant | 100 |

RCT, randomized controlled trial.

Other phrases were much less linked to 0.05 to 0.15 *P* values, namely "a significant trend," "all but significant," "an increasing trend," and "nominally significant" (**Fig 5**). Similar differences were found for less frequent phrases, with some strongly connected to *P* values just above 0.05 (**S2 Fig**).

## Discussion

### Principal findings

This study systematically assessed more than half a million full-text publications of RCT published between 1990 and 2020 for the prevalence of specific phrases linked to almost but formal nonsignificant reporting (i.e., *P* values just above the 0.05 threshold), including temporal trends and manual validation of the associated *P* values. We present an estimate of 9% of RCTs using specific phrases to report *P* values above 0.05. This prevalence has remained relatively stable in the past 3 decades. We also determined fluctuations over time in the frequently used nonsignificant phrases. Some phrases gained popularity over time, whereas others are more in

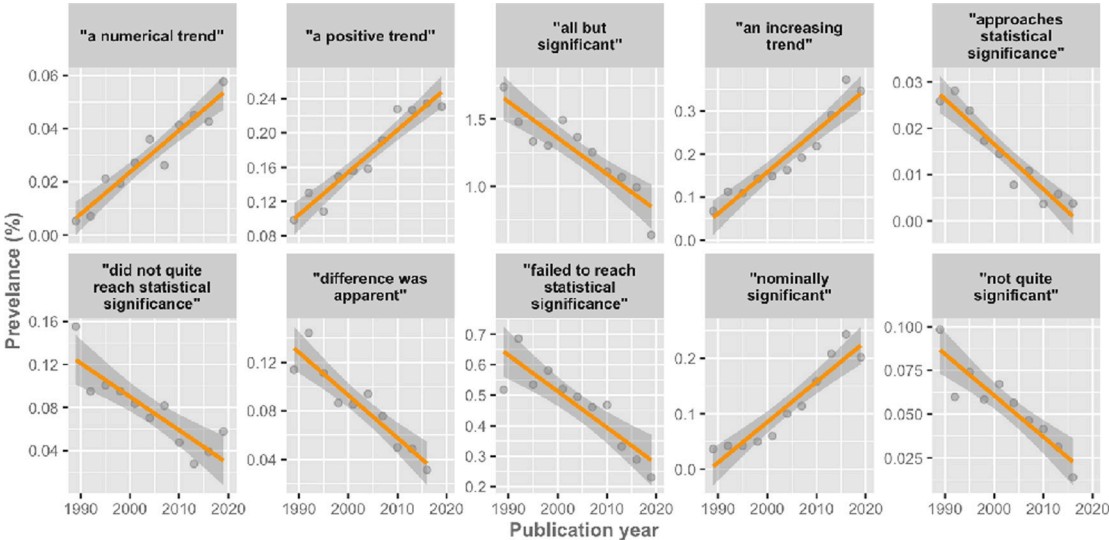

**Fig 3. Temporal plots for phrases with "decisive" evidence (i.e., BFs > 100) for temporal change.** Prevalence estimates are shown as dots, together with the linear regression model fit and corresponding uncertainty. The data can be found here: https://github.com/wmotte/almost_significant/tree/main/Fig_and_Data. BF, Bayes factor.

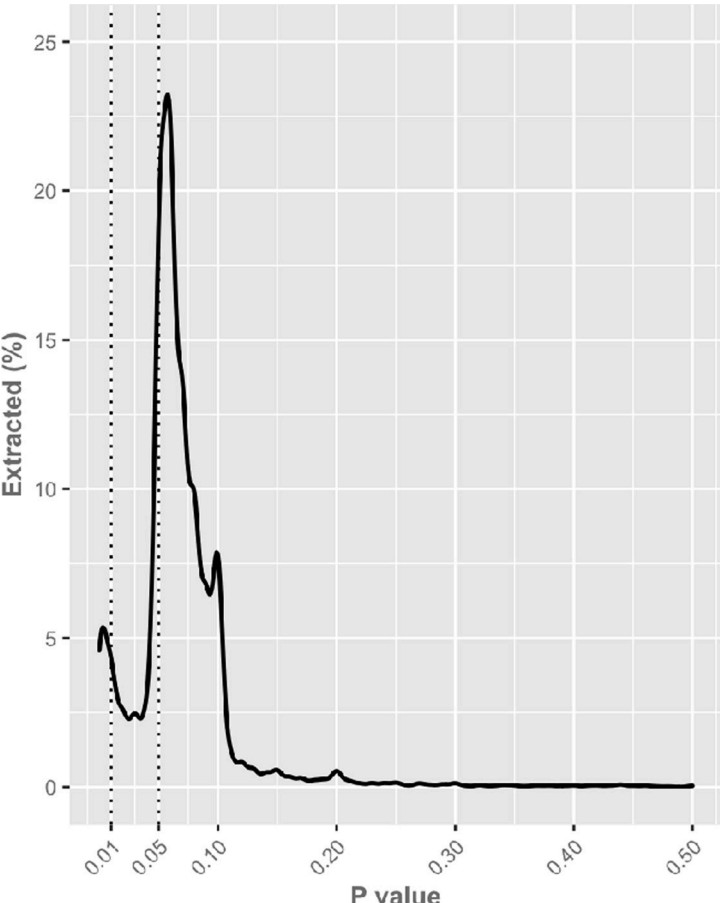

**Fig 4. Density plot of the 11,926 manually extracted *P* values.** The data can be found here: https://github.com/wmotte/almost_significant/tree/main/Fig_and_Data.

decline. Our manual analysis confirmed that most of the phrases described nonsignificant results corresponded with *P* values in the range of 0.05 to 0.15.

## Strengths and limitations

This is the first study to explore a vast body of PubMed-indexed RCTs on the occurrence of phrases reporting nonsignificant results. Given the relatively low frequency of several phrases, such a large sample is essential to effectively quantify prevalence and changes in phrasing over time. Moreover, we also quantified the actual *P* values of the most frequently used phrases reporting nonsignificant results.

Our study also has inherent limitations. First, we predefined more than 500 phrases denoting results that do not reach formal statistical significance. We may have missed phrases with similar meanings. This would lead to an underestimated overall prevalence. However, we did not implement an elastic search strategy in our pdfs, as this could potentially change the interpretation, for example, by removing the negation. We are certain that a specific spin-like phrase is written in the trial report with exact string matching. However, this may have led to underreporting, and the actual prevalence of these phrases may be an underestimation. Second, not all phrases are equally specific in their association with *P* values just above 0.05. Third, we studied English language RCTs only. Generalizations to other languages can

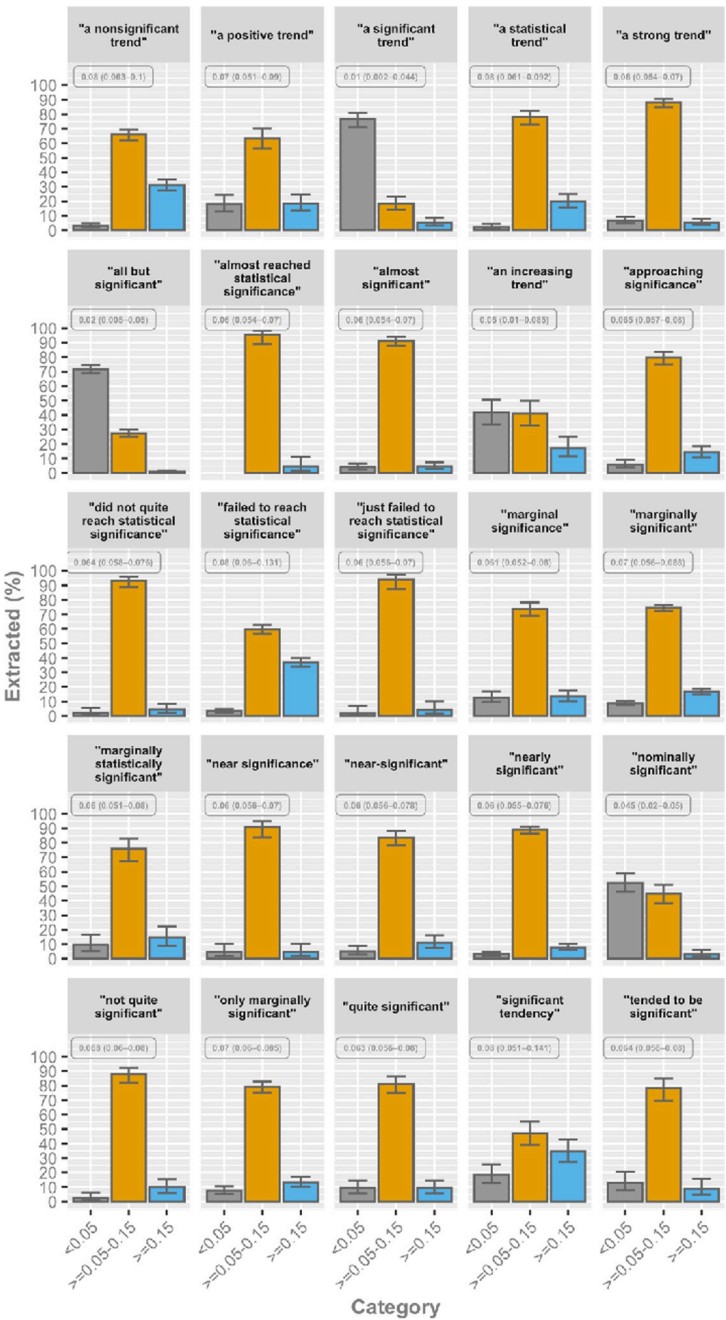

**Fig 5. Category percentages for the 20 most frequent phrases describing nonsignificant results, with at least 100 manually extracted *P* values.** Error bars represent the 95% CI. The associated median *P* value (with the 25% and 75% quantiles) is presented in the upper left corner of each phrase. The data can be found here: https://github.com/wmotte/almost_significant/tree/main/Fig_and_Data. CI, confidence interval.

therefore not be made. Fourth, we only had access to published full texts. This prevents us from drawing causal conclusions as nonpublished manuscripts with specific nonsignificant phrases, which did not undergo a peer-review process, are not available. Connected to that, despite our data collection in September 2020, we missed a relatively large proportion of RCTs published in 2020, rendering our results less stable for the last year. Fifth, we only

characterized *P* values in the direct vicinity of the phrases. Long-range referrals in the text or tables were not included. The association frequencies may hence be conservatively low. Sixth, it remains unknown whether some trials may have had nonsignificant results and used different sentences to describe these results. This may have caused underestimating the prevalence of these types of sentences. Seventh, we do not know whether the *P* value and the corresponding significance phrase actually referred to the study's primary outcomes or whether it described less important secondary or tertiary outcomes. Finally, not all predetermined sentences actually represent a *P* value above 0.05. (e.g., '"marginally significant"). However, we hardly found *P* values lower than 0.05 corresponding with specific phrases in the manual analysis (see S3 Table). For example, the phrase "failed to reach statistically significant results" highlights a fact, although not as neutral as simply stating "nonsignificant results." Therefore, the amount of spin may vary between phrases and potentially overreport some of our individual phrase prevalence estimations that describe marginally significant results.

## Interpretation

Our findings suggest that specific phrasing to report nonsignificant findings remain fairly common in RCTs. RCTs are time- and energy-consuming endeavors, and an "almost significant" result, can, therefore, be a disappointing experience in terms of the interpretation and publication of the results: Did the RCT "find" an effect or not? Our description of the characteristics of the most prevalent phrases can help readers, peer reviewers, and editors to detect potential spin in manuscripts that overstate or incorrectly interpret their nonsignificant results. Our results also support the notion that some phrases are becoming more popular.

The detected *P* value distributions are important in light of the recent discussions to lower the default *P* value threshold to 0.005 to improve the validity and reproducibility of novel scientific findings [1]. *P* values near 0.05 are highly dependent on sample size and generally provide weak evidence for the alternative hypothesis. This threshold can consequently lead to high probabilities of false-positive reporting or P-hacking in clinical trials [22]. However, replacing the common 0.05 threshold with an even lower arbitrary value is not a definitive solution. Clinical research is diverse, and redefining the term "statistical significance" to even less likely outputs will probably have negative consequences. Lakens and colleagues [23] therefore suggest that we should abandon a universal cutoff value and associated "statistical significance" phrasing and allow scholars to judge the clinical relevance of RCT results on a case-by-case basis. Based on our data, we think that such a personalized approach is beneficial for everyone—especially since it is currently unknown if *P* value cutoffs as low as 0.005 do indeed lead to lower false-positive reporting and will lead to more rigorous clinical evidence. A stricter threshold requires large sample sizes in replication studies—which are hardly conducted—and will probably increase the risk of presenting underpowered clinical results.

Moreover, since it is estimated that half of the results of clinical trials are never published [24], mainly due to negative findings, lowering the *P* value threshold may result in more "negative" studies that remain largely unpublished. Although the detrimental effects of lowering the threshold for statistical significance for medical intervention data are disputed [25–27], a recent retrospective RCT investigation showed that shifting the threshold of statistical significance from *P* value < 0.05 to < 0.005 would have limited effects on medical intervention recommendations as 85% of recommended interventions showed *P* values below 0.005 for their primary outcome [28].We are also aware that his will come with new problems and ways to game this new artificial statistical threshold. We think that if authors discuss and judge their threshold value transparently and show the clinical relevance, there is no need to tie oneself to a universal *P* value cutoff. Journal editors and (statistical) reviewers can play an important role

in propagating ideas from the so-called "new statistics" strategy, which aims to switch from null hypothesis significance testing to using effect sizes and cumulation of evidence to explore and determine potential clinical results relevance [29–31]. Chavalarias and colleagues [32] describe in their paper results that are related to the reporting of pv values, effect sizes, and CIg; in the vast majority (88%) of the included RCTs, they found the reporting of a *P* value <0.05. They also highlight that in 2% to 3% of the analyzed abstracts, they found the reporting of CIs and 22% of the abstracts described effect sizes. Despite this improvement, we remain skeptical whether this will not shift the problem and stimulate researchers to overly report their effect sizes and CIs.

Some argue that BFs should replace the quest for statistical significance. Some phrases were associated with BFs that represent "decisive evidence" for temporal changes in our analysis. It is worth mentioning that BFs are considered a good alternative for statistical significance. However, the BFs may be subject to other biases and linguistic persuasion and should be interpreted in light of their research context [33], so this not be a definitive solution.

Our study questions the current emphasis on a fixed *P* value cutoff in interpreting and publishing RCT results. Besides abandoning a universally held and fixed statistical significance threshold, an additional solution may be the 2-step submission process that has gained popularity in the past years [34,35]. This entails that an author first submits a version including the introduction and methods. Based on the reviews of this submission, a journal provisionally accepts the manuscript. When the data are collected, the authors can finalize their paper with the results and interpretation, knowing that it is already accepted.

In conclusion, too much focus on formal statistical significance cutoffs hinders responsible interpretation of RCT results. It may increase the risk for misinterpretation and selective publication, particularly when *P* values approach but do not cross the 0.05 threshold. Fifteen years of advocacy to shift away from null hypothesis testing has not yet fully materialized in RCT publications. We hope this study will stimulate researchers to put their creativity to good use in scientific research and abandon a narrow focus on fixed statistical thresholds but rather judge statistical differences in RCTs on its effect size and clinical merits.

## Supporting information

**S1 Fig. Temporal plots for phrases with "strong" evidence (i.e., BFs between 10 and 100) for temporal change.** Prevalence estimates are shown as dots, together with the linear regression model fit and 95% CI. The data can be found here: https://github.com/wmotte/almost_significant/tree/main/Fig_and_Data. BF, Bayes factor; CI, confidence interval. (DOCX)

**S2 Fig. Category percentages for the phrases describing nonsignificant results with the number of manually extracted P values with occurrences between 30 and 100 times in our manual analysis.** Error bars represent the proportional 95% CI. The associated median *P* value is presented in the upper left corner of each phrase. CI, confidence interval. (DOCX)

**S1 Table. The 505 predefined phrases associated with reporting nonsignificant results.** (DOCX)

**S2 Table. The evidence of temporal change in the phrases with at least 5 time points expressed as the BF relative to no temporal change (lower threshold set to 2.0).** The colors represent the strength of evidence as specified in the main text. BF, Bayes factor. (DOCX)

**S3 Table. All extracted P values within the 3 range categories, as the proportion of the total of 11,926 extractions.**
(DOCX)

## Author Contributions

**Conceptualization:** Willem M. Otte, Christiaan H. Vinkers, Joeri K. Tijdink.

**Data curation:** Willem M. Otte.

**Formal analysis:** Willem M. Otte, Philippe C. Habets, David G. P. van IJzendoorn, Joeri K. Tijdink.

**Investigation:** Willem M. Otte, Christiaan H. Vinkers, Philippe C. Habets, David G. P. van IJzendoorn.

**Methodology:** Willem M. Otte, Christiaan H. Vinkers, Philippe C. Habets, David G. P. van IJzendoorn, Joeri K. Tijdink.

**Project administration:** Joeri K. Tijdink.

**Resources:** Willem M. Otte.

**Software:** Willem M. Otte.

**Supervision:** Christiaan H. Vinkers, Joeri K. Tijdink.

**Validation:** Joeri K. Tijdink.

**Writing – original draft:** Willem M. Otte, Christiaan H. Vinkers, Joeri K. Tijdink.

**Writing – review & editing:** Willem M. Otte, Christiaan H. Vinkers, Philippe C. Habets, David G. P. van IJzendoorn, Joeri K. Tijdink.

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
