## [Editor Report · Decision Letter 0]

30 Apr 2021

Dear Dr Tijdink, 

Thank you for submitting your manuscript entitled "Almost significant: trends and P values in the use of phrases describing marginally significant results in 567,758 randomized controlled trials published between 1990 and 2020" for consideration as a Meta-Research Article by PLOS Biology.

Your manuscript has now been evaluated by the PLOS Biology editorial staff, and I'm writing to let you know that we would like to send your submission out for external peer review.

Please re-submit your manuscript within two working days, i.e. by May 04 2021 11:59PM.

Kind regards,

Roli Roberts

Roland Roberts

Senior Editor

PLOS Biology

rroberts@plos.org

---

## [Decision Letter · Decision Letter 1]

18 Jun 2021

Dear Dr Tijdink,

Thank you very much for submitting your manuscript "Almost significant: trends and P values in the use of phrases describing marginally significant results in 567,758 randomized controlled trials published between 1990 and 2020" for consideration as a Meta-Research Article at PLOS Biology. Your manuscript has been evaluated by the PLOS Biology editors, an Academic Editor with relevant expertise, and by three independent reviewers.

You’ll see that the reviewers are broadly positive about your study. Reviewer #1’s requests are mostly for clarification. Reviewer #2's requests involve some re-framing, especially around what question you set out to address, and your remedial recommendations. Reviewer #3 asks that you compare their results with those of a 2016 JAMA paper that you cite, and has a number of requests for methodological detail.

In light of the reviews (below), we will not be able to accept the current version of the manuscript, but we would welcome re-submission of a much-revised version that takes into account the reviewers' comments. We cannot make any decision about publication until we have seen the revised manuscript and your response to the reviewers' comments. Your revised manuscript is also likely to be sent for further evaluation by the reviewers.

We expect to receive your revised manuscript within 3 months. 

**IMPORTANT - SUBMITTING YOUR REVISION**

*Re-submission Checklist*

*Published Peer Review*

*PLOS Data Policy*

*Blot and Gel Data Policy*

Sincerely,

Roli Roberts

Roland Roberts

Senior Editor

PLOS Biology

rroberts@plos.org

REVIEWERS' COMMENTS:

Reviewer #1:

[identifies herself as Agnes Dechartres]

Review Plos Biology PBIOLOGY-D-21-01082R1, « Almost significant : trends and p values in the use of phrases describing marginally significant results in 567,758 randomized controlled trials published between 1990 and 2020 »

This is a research on reseach report evaluating the prevalence of common phrases associated with non significant results in 567,758 randomized controlled trials (RCTs) published between 1990 and 2020, the trend over time and the associated p-values in a subset of RCTS.

The topic is interesting and fits well the meta-research section of Plos Biology. The manuscript is overall well-written.

Please find below my comments. I hope you will find them useful.

Major comments :

- The reference and link with spin should be made more apparent

- We need more information on the methods section on how RCTs were identified (were they all those indexed as randomized controlled trials in publication type ?), how did the authors exclude those that were not actual RCT reports ?

- How did the authors processed technically to search for the common phrases associated with non-significant results ?

- The authors evaluated as a second step, p-values following the phrases commonly associated with a non-significant result in a subsample of 29,000 RCTs. Why 29,000 ? as they did that quite automatically, was it not possible to do it for the whole sample ?

- There is a contradiction in the Discussion section of the manuscript between what the authors did and showed : the prevalence of common phrases associated with non-significant results, mainly with a p-value between 0.05 and 0.15 is relatively low but still present (and this should probably be more highlighted and discussed, the prevalence seems low but it concerns all RCTs and many of them may be positive so one limitation is that they did not give the prevalence of such common phrases among RCTs with non-significant results) and seem steady over time. And the implications they highlight and the conclusion they reach: the abandon of formal statistical significance cutoff. The conclusion should be more in line with the results. 

- The authors mentioned in the discussion that it was previously discussed to lower the default p value threshold to 0.005. I think it was mainly proposed for pre-clinical research. Do the authors think it is applicable for clinical research including clinical trials ? 

Minor comments :

- Title : I suggest rephrasing the title as the current version is not completely clear something like : 

« Almost significant » : Prevalence and trend of common phrases associated with non-significant results in 567,758 randomized controlled trials published between 1990 and 2020

- Abstract : the authors state that « The question is how non-significant outcomes are reported in RCTs over the last thirty years. » They should be careful in wording as this is not what they evaluated here. They only evaluated the prevalence of common phrases associated with non-significant results in RCTs (and not specifically in non-significant ones). To answer the first question, they should have first identified a sample of RCTs with non-significant results and evaluated how they were reported

- Abstract : This should be made clear in the abstract that the authors extracted associated p-values in a subsample of RCTS as the sentence ending with « in 567,758 RCT full texts between 1990 and 2020 and manually extracted associated p values » may be misleading

- Abstract : rephrase the sentence « Phrase data was modelled with Bayesian linear regression » to something like : temporal trend in phrase data was modelled with Bayesian linear regression

- Abstract : the conclusion should be more directly linked to the results 

- One of the phrases considered as over-interpretating non significant results is « failed to reach statistically significant results. » This phrase for me reflects the truth regarding achieving or not the significance threshold and I am not sure that it corresponds to an over-interpretation

- The introduction is a bit long and sometimes a bit pedantic. Some sentences are vague and not really clear. Among these, for example « Individual clinical researchers are subject to regulations, traditions and procedures… ». 

- In the introduction, the authors mention two previous meta-analytic studies showing that positive reporting and interpretation of primary outcomes in RCTs were frequently based on non-significant results. This sentence should be corrected as these studies were not meta-analyses. In addition, the sentence is misleading as these studies evaluated the prevalence of spin (misleading language) in RCTs with a non-significant primary outcome. 

- Methods section : The sentence page 6, starting with « To increase the robustness of individual phrase prevalence estimations we binned… »

- A flow chart would be helpful

- In the results section, the sentence page 10 starting with interrater p value variability… « is not really clear. This was not annonced in the methods

- Discussion section : sentence at the end of page 14 ending with « irrespective of the publication status ». I think this is rather the opposite : With that process, publication does not depend on study results

Reviewer #2:

The manuscript reports on a meta-research study on how authors of reports on randomized controlled trials (RCTs) use specific wordings suggesting "marginally significant" results. The number of articles analyzed is impressive (grossly 570,000, more than 80% of all published RCTs that can be found in PubMed). But automatic analysis of such a large number also makes it difficult to attain a fine granularity, and this raises many issues. This is only imperfectly mitigated by manual extraction of P-values corresponding to a random sample of 29,000 extracted phrases. On the positive side, I must acknowledge that this, also, represents an enormous amount of work.

Major comments

1. Overall, the manuscript is unclear on what problem is precisely studied. Mostly the issue would be related to the use of threshold P-values for statistical testing. But actually, the issue is broader, and this is imperfectly tackled in the manuscript. The suggestion to lower the P-value threshold from 0.05 to 0.005, as proposed by John Ioannidis, is discussed in terms of lowering false positive reporting, but this is actually a completely different issue. Undoubtedly, authors would develop strategies to describe a P-value of 0.0051as 'marginally significant', a 'trend to significance', etc. In a similar way, thresholds for Bayes factors are presented in the manuscript. It is mentioned that there is no evidence that Bayes factors would be reported with suspicious phrasing, but one may argue that Bayes factors there is no evidence the Bayes factors are reported in RCTs at all (despite this being advocated, references should be given in the discussion when coming to this). Apart from an easy joke, if RCTs were routinely reported with Bayes factors and the thresholds given almost universally endorsed, then—again—authors may begin a 'significance dance' to advertise a Bayes factor of 3.18. The discussion also advocates the use of effect sizes and confidence intervals. But this is another issue, also. How many of the reports analyzed also reported measures of effect size and a confidence interval? Perhaps a vast majority of them, even those using suspicious phrasing. Moreover, a 95% confidence interval also somehow emphasizes a significance level at 0.05. Of course, it brings much more information. But how many readers will look whether the 95% CI of the relative risk includes 1 or not? All these issues are mixed up in the manuscript, and it would benefit from better delineating precisely the underlying problems. It seems to me that they are primarily caused by the use of thresholds for decision, whatever they are. But the authors may have a different view. It should simply be clarified.

2. As a follow-up comment, the topic may be narrowed to how authors phrase results close to, but above, the significance threshold, or enlarge it to envision what could be done to solve the issue (if it is really problematic). Currently, the manuscript is quite in-between. But the larger vision would raise many other issues: how do we dimension a study if there is no decision rule? (the power calculation would not apply) When should be stop a study without decision rules? Do we need a clear-cut conclusion of a RCT? (I would tend to say "no", but other authors have done sensible comments on that, and this relates to the suggestion to judge clinical relevance of RCTs on a case-by-case basis in the discussion; however on that specific point, I would tend to consider that this is still the case, whatever wording the authors use to describe their results). Bayes factors are used here, and are interesting. But perhaps a good option may be to analyze RCTs in a Bayesian framework, and report posterior probabilities for decision making (again one may differentiate between decision-making for a particular trial, i.e. stopping recruitment or not, and decision-making about the drug or intervention evaluated). Other tools have also been proposed to improve the interpretation of RCTs (see e.g. Shakespeare Lancet 2001; 357: 1349-53).

3. Last on this broad issue, the topic of p-hacking could also have been covered, and there have been large-scale analyses of the distribution of P-values on that topic, too (e.g. Adda et al. PNAS 2020).

4. The end of the introduction suggests the need to study how results with P-values just above 0.05 are described in RCTs, but the last paragraph and the study does it the other way round (what are the P-values for predefined 'non-significance-related phrases'), and that way, the question looks less pertinent. 

5. A huge work has been done to search a predetermined set of phrases reportedly associated with 'non-significant' results. But actually, there is no good evidence that those phrases indicated a 'significance dance' from the authors. Why not describe a result with P=0.048 (assuming a 0.05 predefined threshold) as 'marginally significant'? And actually, the figure 3 shows P-values as low as 0.01, and the figure 4 shows some sentences with a fair proportion of P<0.05. So all the phrases do not correspond to 'almost significant' results. I acknowledge that the peak at 0.05-0.10 is impressive, but the text should better reflect that all those statements did not correspond to P-values > 0.05.

6. I was surprised that only 272 of the 505 predefined phrases were encountered in half a million RCT reports, since those sentence were encountered in the literature. The blog apparently also accounts for the psychology literature and non-randomized studies, but this was striking. Also, I failed to access the URL given for that blog. It should therefore be checked for accurateness.

7. I wondered whether sentences such as 'did not quite reach statistical significance' actually correctly interpret the P-value as being above the significance threshold. I am not a native English speaker, so that I may miss the exact meaning of the sentence, but looking at such a sentence, I would understand 'P > 0.05' but perhaps not very high. This sounds different from 'marginally significant' or 'borderline significant' or 'almost significant'. On the sentences, also, some also seem to me to indicate statistical significance. For instance I would interpret 'nominally significant' as 'P<0.05, but close too'. If the study had extracted P-values between, say, 0.05 and 0.20 and than looked at how they were described, the story would be different. But it seems that all sentences searched were not equally suggesting a spin towards significance when P>0.05. This is discussed (second limitation), but too shortly.

8. The P-value threshold at 0.05 is most common. But some trials may have predefined other thresholds. At least for the RCTs for which P-values were manually extracted, did the authors look for such information?

9. On what basis was it decided to manually extract P-values for 29,000 phrases?

10. The very end of the abstract advocates the reporting of exact P-values. This is however not directly related to the study results. And actually, it seems that it was possible to extract the exact P-values corresponding to many extracted phrases. So, they were actually reported.

11. Another important issue is how to interpret secondary outcomes of RCTs. Actually, many argue that, in the absence of correction for multiplicity or gatekeeping approach, one should not look at the P-values for secondary outcomes. Moreover, although this is unfortunately not always the case, the conclusion of a RCT on efficacy should target the primary outcome. It would therefore be interesting to distinguish primary and secondary outcomes here. I however understand that this would be virtually impossible to automatize.

Additional comments

1. The term 'phrase model' is unclear.

2. The introduction refers to 'low evidence results'. Yet, as the authors later acknowledge, the information brought by a study is not so different if it achieved P=0.049 or P=0.051. Moreover, many other elements may affect the level of evidence, or confidence in the evidence brought by the study, such as methodological characteristics, risk of bias, etc. This should be reflected in the sentence, or rephrased.

3. How was the density transformed to a percentage on the figure 3?

4. I wondered what 95% 'proportional' confidence intervals were (legend of figures 1 and 4). Are they the confidence interval of the proportion? Is that a common wording? (I never saw that).

Reviewer #3: 

The authors aimed to assess the prevalence of a specific set of phrases describing non-statistically significant results in articles reporting the findings of RCTs. They used exact matching to identify instances of this corpus across 567,758 articles. I have the following comments to improve the reporting and conclusions:

1) The authors cited the work of David Chavalarias and colleagues (JAMA 2016) but they did not compare their findings to theirs. In particular, the sample of Chavalarias et al included RCTs as indexed by Medline and they already quantified the evolution over time of the prevalence of p value <0.05 (thus of p values >= 0.05). They already found that the proportion of p values <0.05 was 91.4% in RCTS (completely consistent with the findings of the authors of ~9%) and they also found that this proportion remained constant over time. Chavalarias et al. also assessed the statements associated to p values in a subsample.

2) The premise of the article is that describing RCT results as 'almost significant' (or 'marginally significant) can influence the perception of readers. The findings show that 'almost significant' is not frequent (only 0.14%). 'Marginally significant' was more prevalent (1.36%) but still much smaller than the prevalence of p values > 0.05 (about 9%). In addition, the use of these phrases seem to be stable over time (in particular the phrases do not appear among increasing or decreasing trends in Figure 4). The conclusions of the authors do not seem to be consistent with these findings. 

3) The Methods section does not provide any detail regarding the identification of RCTs. Did the authors use a filter to query PubMed? Which one? See 10.5195/jmla.2020.912 What is the sensitivity and specificity of the filter? How did the authors address the false positives? 

4) The authors could not retrieve 18.53% of identified articles. What was this proportion across publication years? Did it change? Were the authors less likely to retrieve full-text articles of older articles? What is the implication on their findings?

5) The URL provided in reference 11 is not accessible. I find it very problematic. Is the list of phrases from Academia Obscura including all 505 phrases used by the authors? How was this list developed? Is this corpus derived specifically from RCTs? Is it based on articles spanning from 1990 to 2020? In addition, how can the reader understand the extent to which these phrases are sensitive and specific to the reporting of non-significant results? The authors refer to "potentially associated with reporting non-significant results" but the reader has no way to make a judgment about the meaning of "potentially". 

6) The Methods section does not mention that the authors selected articles published in English. The proportion of excluded articles based on language is not reported. 

7) 233 (46%) phrases were not identified among the 567,758 pdf. What were the main differences between the 272 phrases found at least once and the 233 phrases? 

8) The authors searched for exact matches. They should justify why they did not use approximate string matching methods? What is the implication on their findings?

9) The authors grouped selected articles into 3-year periods. The first period would be 1990-1992 and the last period would be 2017-2019. How did they handle the last period from Jan to Sep 2020?

10) There is no detail regarding the method for sampling articles to manually extract p values. Was random sampling stratified according to publication year or other factors? 

11) Did the authors searched for exact matches in both the abstract and the main body of the selected articles? Were results different when stratified by location (abstract vs main text)?

12) Figure 4 shows large percentages of p values <0.05 for certain phrases. Across all extracted p values, what is the frequency and percentage of p values <0.05? What is the explanation? Is it that the significant threshold was lower than 0.05 in these papers? Was it that the selected phrases are not specific enough? It seems to be a considerable limitation in face of the objective of the authors to assess reporting of non-statistically significant findings. 

13) In the Discussion, the authors conclude: "We present a robust estimate of nine percent of RCTs using specific language to report P values around 0.06." I suggest rephrasing. Because of the lack of information regarding the Methods and the limitations mentioned above, I find the claim that the estimate is robust undue. And I am not following 'p values around 0.06'. 

14) In the abstract, the authors state "the phrase prevalence remained stable over time, despite all efforts to change the focus from P < 0.05 to reporting effect sizes and corresponding confidence intervals" Did they assess if effect sizes and confidence intervals were reported alongside non statistically significant results? If not, I suggest modifying this conclusion.

---

## [Decision Letter · Decision Letter 2]

22 Dec 2021

Dear Dr Tijdink,

Thank you for submitting your revised Meta-Research Article entitled "Almost significant: Prevalence and trends of common phrases associated with marginally significant results in 567,758 randomized controlled trials published between 1990 and 2020" for publication in PLOS Biology. I have now obtained advice from the original reviewers and have discussed their comments with the Academic Editor. 

Based on the reviews, we will probably accept this manuscript for publication, provided you satisfactorily address the remaining points raised by the reviewers. Please also make sure to address the following data and other policy-related requests.

IMPORTANT:

a) Please can you change your Title to "Analysis of 567,758 randomized controlled trials published over 30 years reveals trends in phrases used to discuss results that are not statistically significant" or "...that do not reach statistical significance" or "...that are only marginally significant," whichever is most accurate.

b) Please address the remaining concerns raised by the reviewers. While reviewer #3 remains critical, after discussion with the Academic Editor, we think that their concerns can be addressed by clarifying methods, toning down claims and making the limitations more explicit.

c) Please could you supply a blurb, according to the instructions in the submission form?

d) Please check that you adhere to our guidelines pertaining to systematic reviews and meta-analyses (e.g. by including a completed PRISMA checklist and flow diagram) https://journals.plos.org/plosbiology/s/best-practices-in-research-reporting#loc-reporting-guidelines-for-specific-study-types

e) Your current financial declaration says “The author(s) received no specific funding for this work.” Please can you confirm that this is correct.

f) Please address my Data Policy requests below; specifically, while we recognise that your raw data and code are presented in your GitHub deposition, we need you to supply the numerical values underlying Figs 2ABC,3, 4, 5, S1, S2. Please cite the location of the data clearly in each relevant Fig legend.

We expect to receive your revised manuscript within three weeks. 

*Published Peer Review History*

*Early Version*

Sincerely,

Roli Roberts

Senior Editor,

rroberts@plos.org,

PLOS Biology

DATA POLICY:

Many thanks for providing the raw data and code in Github. However, we also need the individual numerical values that underlie the Figures of your paper be made available in one of the following forms:

Regardless of the method selected, please ensure that you provide the individual numerical values that underlie the summary data displayed in the following figure panels as they are essential for readers to assess your analysis and to reproduce it: Figs 2ABC,3, 4, 5, S1, S2. NOTE: the numerical data provided should include all replicates AND the way in which the plotted mean and errors were derived (it should not present only the mean/average values).

DATA NOT SHOWN?

REVIEWERS' COMMENTS:

Reviewer #1:

Review PBIOLOGY-D-21-01082R2 manuscript entitled "Almost significant: Prevalence and trends of common phrases associated with marginally significant results in 567,758 randomized controlled trials published between 1990 and 2020"

I would like to thank the authors for having answered my comments and modified their manuscript accordingly.

I have some last comments:

- Abstract section: it is not clear what represents the rate 68.1% at the end of results presented. Is it the % of p-values between 0.05 and 0.15? Please clarify

- Abstract section: in the conclusion, "demonstrate" is too strong. Please consider using "show" instead

- The introduction is still very very long and I still find that some sentences or wording are too vague or unclear. For example: "The use of this fixed threshold has important disadvantages as the context determines the corresponding threshold". I am not sure that the catch-22 situation expression is clear for most researchers. Also not really clear "insufficient evidence results (including factors such as high risk of bias, other methodological weaknesses", "given the relatively rules-oriented RCT research environment", "the success of an RCT" 

- I am happy with the reference to spin in the introduction but I think it is not necessary to provide a catalog of findings from recent studies on that topic

- I also find that the plan of the introduction is not easy to follow. I really suggest to condense and to be more factual following a clear-cut plan

- I would be more careful in the formulation of the objectives in particular the part "explore what types of phrases are used when reporting statistical non-significant findings". It is not exactly what the authors did. This formulation suggests that they searched first for non-statistically significant results and then look at the sentences around. In this study, they did the contrary: they look the p-values following a sentence that could be related to a marginally significant result.

- I think this is also a limitation to acknowledge more clearly: having focused on all RCTs and not only on those with non-statistically significant results

- In the methods section, why did the authors use "clinical trial" as publication type and not "randomized controlled trial"?

- Typo in the paragraph statistical analysis: principled

- Another sentence vague and pedantic in the methods: "with the tendency of humans to understand the world by applying thresholds to continuous spectra"

- In the results: the sentence "the overall phrase-positive RCT prevalence was stable over time (8.65%). Does it correspond as indicated to the overall prevalence, ie 49,134/567,758? It looks so but "stable over time" in the same sentence is confusing. I would make two sentences.

- Results, associated p-values: I would expect here the prevalence of p-values in the range 0.05-0.15 and not only the median. Please also make two sentences

- Results, associated p-values: The sentence "the highest percentages for relative frequent phrases were found in this particular range.. " is unclear

- Discussion, in principal findings, "no specific reasons were found for these differences over time". For me, it is not really a finding as they did not report to study that. I would eventually discuss that point later in the discussion

- The conclusion should be more in line with the results. In particular, the sentence "fifteen years of advocacy to shift away from null hypothesis testing has not yet fully materialized in RCT publications"

- The flow chart could be improved

- Figure 5: I would add Q1-Q3 with the median if possible 

Reviewer #2:

I thank the authors for their answers and revision of the manuscript.

Some issues may still be considered:

1. When describing the "mythical paradigm" of statistical testing in the introduction, authors may be clear that the statistical framework does not mandate that P-values < 0.05 indicate a true effect and those >0.05 indicate no effect. This is simply a rule to reject the null or not, based on long-run properties allowing to control the type I and II (provided studies are correctly powered) error rates. So it is more the interpretation that has pervasively permeated the medical community that has led to the shortcut P<0.05 = true effect and P>0.05 = no effect. This may be clarified.

2. The flow chart should have arrows going out with the "excluded" studies at each step, and the reason why they would go out of the flow. For instance, when starting from the 696,842, 129,084 are excluded before full test download: they should appear on the chart, with the reason why full text was not retrieved (there could be several reasons for different sets of studies).

3. The sixth limitation is quite vague, and should be made more precise. I believe two reviewers alluded to this. The way the data were analyzed did not look at whether the RCTs achieved their primary outcome or not, nor if the sentences and p-values extracted related to the primary outcome. So the limitation should be clarified: we do not know if what is reported relates to the primary outcome or not, and there is no stratification according to whether the trials reached their primary endpoint (showed a "significant result" of their primary outcome) or not.

Some minor points

1. Abstract, "RCTs regularly use specific phrases": RCTs themselves do not use any sort of sentence. Those phrases can be found in their reports. I would therefore write it differently, such as "Our results show that RCTs reports regularly contain specific phrases describing …" (not sure either there would be a need for a real demonstration).

2. In the introduction, I would not allude to the exact number of full-texts analyzed, since this is already part of results, but more on the strategy: "We do this by quantitatively analysing 567,758 RCT full-texts …" would be changed to "We do this by quantitatively analysing RCT full-texts in English registered in the last decade …"

3. What is the need for "direct" in the sentence "To obtain direct evidence on phrase changes …"? Would another type of model produce indirect evidence? Moreover, the evidence itself would be more in the data than in the model. 

4. The figure 1 should be referenced in the text, e.g. "We obtained the full text of 567,758 full-texts of the total of 696,842 PubMed-registered RCTs (81.47%) (figure 1)".

5. The seventh limitation (related to not using elastic search) could be put earlier, just after the fact that a fixed set of phrases was used: such a fixed set was used, and exact match was searched. 

Reviewer #3:

The response of the authors and the revised manuscript reveal flaws in the methodology. The methodology chosen by the authors does not allow answering the objective of assessing "how non-significant outcomes are reported in RCTs over the last thirty years" or "how phrases expressing non-significant results are reported in RCTs over the past thirty years". It is impossible to assess the robustness of the approach, but Figure 4 clearly shows that the approach is biased. In addition, these shortcomings are not acknowledged or are downplayed by the authors.

1) The whole methodology relies on a list of 505 phrases compiled and shared through a blog post by Matthew Hankins. The manuscript does not explain how the list was developed. The original blog post only mentions: "The following list is culled from peer-reviewed journal articles in which (a) the authors set themselves the threshold of 0.05 for significance, (b) failed to achieve that threshold value for p and (c) described it in such a way as to make it seem more interesting." So, the internal and external validity of the findings relies completely on a list of phrases developed by a single researcher based on an irreproducible non-systematic review of an ad hoc selection of articles of unspecified nature (biomedical research? RCTs?). As highlighted by all reviewers of this manuscript, the correct method should be to first identified a sample of RCTs with non-significant results and to evaluate how they are reported

2) The authors downplay this major flaw. They state "First, we [Matthew Hankins?] pre-defined more than five hundred phrases denoting results that do not reach formal statistical significance. We may have missed phrases with similar meanings. This would lead to an underestimated overall prevalence." Figure 4 shows that a certain proportion of p values associated with the phrases are below 0.05. It clearly invalidates the approach. It is also incorrect to state that the prevalence is underestimated. The authors had the opportunity to report the exact proportion of p values < 0.05 but they did not. It is difficult to guess the proportion based on Figure 4, a histogram would be better. It is the area under the pdf to the left of 0.05. The proportion of p values <0.05 seems to be at least 2.5%. It is a relatively large percentage compared to the estimated 9%. The authors minimize this finding by stating "Finally, not all predetermined sentences actually represent a p value above 0.05. (e.g. 'marginally significant'). However, in the manual analysis, we hardly found p values lower than 0.05 that corresponded with specific phrases".

3) The conclusion of the article "too much focus on formal statistical significance cut-offs hinders full transparency and increases the risk for misinterpretation and selective publication, particularly when P values approach but do not cross the 0.05 threshold." and of the abstract "To enhance responsible and transparent interpretation of RCT results, researchers, clinicians, reviewers, and editors should abandon the focus on formal statistical significance thresholds and stimulate reporting of P values with corresponding effect sizes and confidence intervals and focus on the clinical relevance of the statistical difference found in RCTs." are not supported by the findings and go well beyond them.

---

## [Editor Report · Decision Letter 3]

31 Jan 2022

Dear Dr Tijdink,

On behalf of my colleagues and the Academic Editor, Isabelle Boutron, I'm pleased to say that we can in principle accept your Meta-Research Article "Analysis of 567,758 randomized controlled trials published over thirty years reveals trends in phrases used to discuss results that do not reach statistical significance" for publication in PLOS Biology, provided you address any remaining formatting and reporting issues. These will be detailed in an email that will follow this letter and that you will usually receive within 2-3 business days, during which time no action is required from you. Please note that we will not be able to formally accept your manuscript and schedule it for publication until you have any requested changes.

PRESS: We frequently collaborate with press offices. If your institution or institutions have a press office, please notify them about your upcoming paper at this point, to enable them to help maximise its impact. If the press office is planning to promote your findings, we would be grateful if they could coordinate with biologypress@plos.org. If you have not yet opted out of the early version process, we ask that you notify us immediately of any press plans so that we may do so on your behalf.

Sincerely,

Roli Roberts

Roland G Roberts, PhD 

Senior Editor 

PLOS Biology

rroberts@plos.org